# Hardness of Low Rank Approximation of Entrywise Transformed Matrix Products

**Tamas Sarlos**
Google Research
stamas@google.com

**Xingyou Song**
Google Deepmind
xingyousong@google.com

**David P. Woodruff**
Carnegie Mellon University
dwoodruf@cs.cmu.edu

**Qiuyi (Richard) Zhang**
Google Deepmind
qiuyiz@google.com

## Abstract

Inspired by fast algorithms in natural language processing, we study low rank approximation in the entrywise transformed setting where we want to find a good rank $k$ approximation to $f(U \cdot V)$, where $U, V^\top \in \mathbb{R}^{n \times r}$ are given, $r = O(\log(n))$, and $f(x)$ is a general scalar function. Previous work in sublinear low rank approximation has shown that if both (1) $U = V^\top$ and (2) $f(x)$ is a PSD kernel function, then there is an $O(nk^{\omega-1})$ time constant relative error approximation algorithm, where $\omega \approx 2.376$ is the exponent of matrix multiplication. We give the first conditional time hardness results for this problem, demonstrating that both conditions (1) and (2) are in fact necessary for getting better than $n^{2-o(1)}$ time for a relative error low rank approximation for a wide class of functions. We give novel reductions from the Strong Exponential Time Hypothesis (SETH) that rely on lower bounding the leverage scores of flat sparse vectors and hold even when the rank of the transformed matrix $f(UV)$ and the target rank are $n^{o(1)}$, and when $U = V^\top$. Furthermore, even when $f(x) = x^p$ is a simple polynomial, we give runtime lower bounds in the case when $U \neq V^\top$ of the form $\Omega(\min(n^{2-o(1)}, \Omega(2^p)))$. Lastly, we demonstrate that our lower bounds are tight by giving an $O(n \cdot \text{poly}(k, 2^p, 1/\epsilon))$ time relative error approximation algorithm and a fast $O(n \cdot \text{poly}(k, p, 1/\epsilon))$ additive error approximation using fast tensor-based sketching. Additionally, since our low rank algorithms rely on matrix-vector product subroutines, our lower bounds extend to show that computing $f(UV)W$, for even a small matrix $W$, requires $\Omega(n^{2-o(1)})$ time.

## 1 Introduction

The central idea behind the classic problem of low rank approximation (LRA) is to approximate a given matrix with a rank $k$ matrix that preserves the important features of the original matrix, while being computationally more efficient and statistically more stable. One particular area of interest in LRA is the low rank decomposition of entrywise transformed matrices, where the entries of the original matrix are transformed through a non-linear function before being approximated, with various applications in kernel methods, self-attention, and likelihood computations [Choromanski et al., 2021, Levy and Goldberg, 2014]. In these settings, the Gram matrix of a dot-product kernel, the attention module's product operator, or even the non-linear computation of activation in a deep network can all be represented as $f(UV)$, where the inputs are low dimensional matrices $U, V$. Note that while $UV$ is low rank, $f(UV)$ may not be, with the rank blowup dependent on the choice of the transformation $f$.

37th Conference on Neural Information Processing Systems (NeurIPS 2023).

For entrywise transformed low rank approximation, we aim to find an approximately optimal rank $k$ approximation, in terms of relative Frobenius norm error, to the matrix $A = f(UV)$ given $U \in \mathbb{R}^{n \times r}$ and $V \in \mathbb{R}^{r \times d}$, where $f$ is a scalar transformation. In this paper, our goal is to study the functions $f$ such that this task is solvable in subquadratic $O(n^{2-\epsilon})$ time and for ease of presentation, we assume $n = d$ in this section. This problem was studied in the distributed setting in Woodruff and Zhong [2016] for functions $f(x) = |x|^p$, where the goal was to minimize communication. It was also studied in the streaming setting, where it was shown that $O(n \operatorname{poly}(k/\epsilon))$ memory suffices to solve rank $k$ approximation with additive error $O(\epsilon \|U\| \|V\|)$ for the function $f(x) = \log(|x| + 1)$ in a single pass [Jiang et al., 2021], improving the earlier work of [Liang et al., 2020]. In a related paper, Han et al. [2020] present an algorithm for low-rank approximation of polynomial entrywise transforms; however, they make no comparison to the optimal rank $k$ error.

In the setting when $U = V^\top$, certain choices of $f(x)$ can surprisingly admit subqradratic relative error algorithms, such as when $f$ represents a positive semi-definite (PSD) kernel [Musco and Musco, 2017]. Moreover, for any PSD matrix of any rank, by sampling according to the leverage scores of the matrix square root of the kernel, recent work shows that relative error low rank decomposition is possible in $O(n(k/\epsilon)^{\omega-1})$ time, where $\omega \approx 2.373$ is the exponent of matrix multiplication; see [Musco and Woodruff, 2017, Bakshi et al., 2020]. Note that in many cases when $U = V^\top$, carefully choosing $f$ will result in $A = f(UV)$ being PSD. In fact, when $u_i$ are all unit norm, if $f$ admits a Taylor expansion with non-negative coefficients, then $A = f(UU^\top)$ is a PSD matrix and admit sub-quadratic low rank approximations.

On the surface, this astounding algorithmic result seems to conflict with subquadratic lower bounds for basic linear algebraic primitives for many kernel functions, especially the exponential kernel used in the attention architecture. Specifically, such work looks at kernel matrix vector products and shows they take $\Omega(n^{2-o(1)})$ time for relative and additive error approximations [Keles et al., 2023]. Furthermore, these bounds can be refined to hold in the restricted regime when the entries of the matrix are $\Omega(\sqrt{\log(n)})$ [Alman and Song, 2023]. These conditional lower bounds, as with others for linear algebraic problems, are derived from hardness based on the Strong Exponential Time Hypothesis (SETH) [Lokshtanov et al., 2013].

The crucial observation to resolve this seeming contradiction is to recognize that the low rank approximation allows for an error bound that depends on the error of the best rank-$k$ approximation, which can be much larger that the error tolerated in the previous lower bounds. Therefore, hardness for LRA is inherently different than hardness for approximating the entire matrix additively or than matrix-vector (MV) multiplication. In fact, we will show that hardness for MV multiplication is, in some sense, strictly easier to establish.

Some related lower bounds include the work of Backurs et al. [2017] that solving kernel Support Vector Machines (SVM), ridge regression, or Principal Component Analysis (PCA) problems to high accuracy or approximating kernel density estimates up to a constant factor for kernels with exponential tails, requires $n^{2-o(1)}$ time assuming SETH. Also, Alman et al. [2020] show that for linear algebraic primitives for the Laplacian of a graph with weights given by a kernel, most operations, such as $\epsilon$-approximate matrix-vector multiplication, are hard, although their hardness results assume a $\log(1/\epsilon)$ dependence when reducing to SETH.

## 1.1  Our Contributions

In the setting of entrywise-transformed low rank approximation, we show that we cannot get subquadratic relative error LRA generally when either 1) $U \neq V^\top$ even for PSD kernel functions or 2) for transformations that are approximately polynomials of $|x|$ even for constant degree. Therefore, without multiple strong structural assumptions on $A$, there is no subquadratic approximation algorithm, even when $r = \Theta(\log(n))$ and the rank of the transformed matrix $f(UV)$ and the target rank are $n^{o(1)}$. We emphasize these novel hardness results hold for a large class of transformations and in fact generalize to hardness for matrix vector multiplication, which can be derived as a corollary, implying that computing $f(UV)z$ also requires $\Omega(n^{2-o(1)})$ time for many $f$ that have not been studied before (see Theorem 3.2). On the positive side, for the polynomial activation $f(x) = x^p$, we provide an $O(n \cdot \operatorname{poly}(r^p, k, 1/\epsilon))$ time algorithm for relative error LRA and $O(n \cdot \operatorname{poly}(p, k, 1/\epsilon))$ time for additive error LRA via fast tensor-based sketches (see Algorithm 2).

| Upper Bounds | | | |
|---|---|---|---|
| Prior Work | Complexity | Matrix Type | Task |
| Musco and Woodruff [2017] | $O(n \cdot \mathrm{poly}(k/\epsilon))$ | PSD, $U = V^\top$ | Relative LRA |
| Bakshi et al. [2020] | $O(n \cdot (k/\epsilon)^{\omega-1})$ | PSD, $U = V^\top$ | Relative LRA |
| **This work** | $O(n \cdot \mathrm{poly}(2^p, k, 1/\epsilon))$ | $f(x) = x^p, U \neq V^\top$ | Relative LRA |
| **This work** | $O(n \cdot \mathrm{poly}(p, k, 1/\epsilon))$ | $f(x) = x^p, U \neq V^\top$ | Additive LRA |
| Lower Bounds | | | |
| Backurs et al. [2017] | $\Omega(n^{2-o(1)})$ | Gaussian, $U = V^\top$ | PCA, Regression |
| Alman et al. [2020] | $\Omega(n^{2-o(1)})$ | Kernel Laplacians $U = V^\top$ | MV product |
| Keles et al. [2023] | $\Omega(n^{2-o(1)})$ | Exponential Kernels, $U = V^\top$ | MV product |
| **This work** | $\Omega(n^{2-o(1)})$ | $f(x) = |x|^p + O(|x|^{p+1}), U = V^{\top\mathbf{1}}$ | Relative LRA, MV |
| **This work** | $\Omega(\min(n^{2-o(1)}, 2^{\Omega(p)}))$ | $f(x) = x^p, U \neq V^\top$ | Relative LRA, MV |

Table 1: Overview of upper bounds for low rank approximation (LRA) and lower bounds for relative linear algebraic primitives, where the dependence on $r$ is omitted. This table illustrates multiple separation results: 1) LRA is strictly easier than matrix vector (MV) products for positive semidefinite kernels, 2) LRA is strictly easier for functions $f(x)$ that are kernels, even when considering low-degree polynomials of $|x|$, 3) LRA is strictly easier when $U = V^\top$ for $f(x) = x^p$ for $p = \Omega(\log(n))$, which has a $2^{\Omega(p)}$ lower bound when $U \neq V^\top$, 4) additive error LRA is strictly easier than relative error LRA. Our work extends to any function $f(x)$ that admits a Taylor series dominated by $|x|^p$ around $x = 0$. For example, this includes $f(x) = \log(|x| + 1) = |x| + O(|x|^2)$.

We note that previous lower bound techniques do not apply for LRA since relative error LRA approximations of $f(UV)$ are possible without approximating a matrix vector product, as shown by the $O(n^{1+o(1)})$ time LRA algorithm for the popular exponential kernel $f(x) = \exp(x)$ by querying a sublinear number of entries of $f(UU^\top)$ [Musco and Woodruff, 2017, Bakshi et al., 2020]. Indeed, our novel lower bounds differ from previous reductions from Orthogonal Vectors Problem (OVP) by explicitly creating LRA instances and exploiting linear algebraic structural properties of the column spaces of a low-rank tensored matrix.

Specifically, our reduction uses an structural property that if there is a pair of input vectors to OVP which are orthogonal, then under mild assumptions, this implies we can find a *sparse* vector in the column span of the low rank approximation of $f(UV)$, where $U, V$ are matrices containing the input vectors to OVP. Also, our algorithm (Algorithm 1) relies on another critical observation that this sparse vector has uniform magnitude in the non-zero entries, each of which corresponds to a vector with an orthogonal vector pair. By exploiting the structure of this vector, we can lower bound the leverage score of each row that corresponds to a non-zero entry of this sparse vector, and we can appeal to fast leverage score computation algorithms to find a small $n^{o(1)}$-size superset of the support of the sparse vector. Finally, we can quickly check which pairs of entries in the superset correspond to vectors in the original OVP problem via a brute-force search, completing the reduction. The precise reduction is a bit more technically involved, as we also need to allow for additive error for our applications.

In the setting when $U \neq V^\top$, we show novel lower bounds in the case when $f(x) = x^p$ is a polynomial kernel function and admits a simple rank $r^p$ decomposition. We show that we cannot do better than the naïve decomposition and prove an $\Omega(\min(n^{2-o(1)}, 2^{\Omega(p)})$ lower bound (see Theorem 3.3). This implies in the setting when $p = \omega(\log(n))$ and $k = n^{o(1)}$, that surprisingly there is a separation between the (1) $U = V^\top$ setting and the (2) $U \neq V^\top$ setting: when $f(x) = x^{2p}$, in the first setting, previous works show that there admits an $O(nk^{\omega-1})$ approximation algorithm; however, in the second setting when $U \neq V^\top$, our lower bounds imply that there cannot exists a truly subquadratic time algorithm. We emphasize that these exponential lower bounds in the degree for the polynomial kernel are the first of their kind to rule out truly subquadratic algorithms for $p = \omega(\log(n))$, even when $f(x) = x^p$ is a simple polynomial.

We also give an $O(n \cdot \mathrm{poly}(r^p/\epsilon))$ time algorithm for relative error approximation, which agrees with our lower bounds (see Theorem 4.1), showing that our lower bounds for the polynomial kernel when $U \neq V^\top$ are in fact tight. In addition, we provide an $O(n \cdot \mathrm{poly}(p, k, 1/\epsilon))$ time algorithm for additive error LRA that avoids the inherent exponential dependence on $p$, highlighting a separation between additive and multiplicative error LRA when $p = \Omega(\log(n))$. Specifically, we can achieve the easier additive error approximation with fast tensor-based sketching matrices (see Theorem 4.2) by applying comparable techniques to an independent subsequent work on polynomial-based transformers, although the main difference is that they do not output a rank $k$ approximation and suffers a worse dependence on $p, 1/\epsilon$ due to their non-negativity guarantees [Kacham et al.,

2023]. We remark that while the polynomial tensor matrix itself can be approximated relatively in $O(n \cdot \mathrm{poly}(p))$ runtime, we emphasize that the polynomial kernel, as a product of two tensor matrices, must incur additive error or suffer $\Omega(2^p)$ runtime for relative error guarantees.

Finally, we observe that we can generalize our lower bounds for LRA by exploiting the fact that our algorithms for fast LRA reduce to matrix vector product! Therefore, our relative error algorithms are reductions that give MV lower bounds, which are automatically derived for a large number of entrywise transformed matrices, even when $U = V^\top$ (see Theorem 4.3). This implies that in some sense, LRA is an easier problem than MV approximation, implying that our lower bounds for LRA are stronger. We summarize the most relevant prior results and our contributions in Table 1.

## 1.2 Related Work to Transformers and Natural Language Processing

A primary downstream application from our work is in the field of natural language, as it is common to compute similarity matrices $f(UV)$ from token embeddings $U, V$. Such is the case for the popular use of Transformers [Vaswani et al., 2017] and their efficient attention variants [Tay et al., 2022]. The standard attention mechanism consists of the operation $\mathrm{softmax}(\frac{UV}{\sqrt{d}})W$, where in our notation, $U, V, W$ are the "query", "key", and "value" matrices respectively. Simplifying and ignoring normalizations, this can be seen as $f(UV)W$ where $f(x) = \exp(x)$. For a given $f$, one can potentially *linearize* the attention mechanism if there exist $U^\star, V^\star$ such that $f(UV) \approx U^\star V^\star$, as then the order of matrix multiplication can be rearranged into $U^\star(V^\star W)$ which allows memory and runtime in $O(ndk)$.

In the unnormalized softmax case where $f(x^\top y) = \exp(x^\top y)$, Choromanski et al. [2021] notes that $\exp(x^\top y) = \mathbb{E}_{\zeta \sim \mathcal{N}(0, \mathbf{I}_d)} \left[ \exp\left( \zeta^\top x - \frac{\|x\|^2}{2} \right) \exp\left( \zeta^\top y - \frac{\|y\|^2}{2} \right) \right]$ which thus allows defining $U^\star, V^\star$ as random feature matrices from sampled $\zeta_1, \ldots, \zeta_k$. Further use of kernel properties to improve the softmax approximation have been introduced in Choromanski et al. [2022], Likhosherstov et al. [2022]. More generally, for $f$ such that $f(x^\top y) = \mathcal{K}(x, y)$ admits a kernel representation, one may consider using variants of Bochner's theorem [Feller, 1968] to provide approximations via random Fourier features [Rahimi and Recht, 2007].

While works such as Tsai et al. [2019], Kacham et al. [2023] have experimented with linear, polynomial, exponential, and RBF kernels, so far the dominant paradigm, termed the class of *Linear Transformers* Katharopoulos et al. [2020], is to conveniently instead consider the reverse case, where one defines a mapping $\phi : \mathbb{R} \to \mathbb{R}$ in order to define the kernel $\mathcal{K}(x, y) = \phi(x)^\top \phi(y)$. Unfortunately, usually this does not lead to a closed-form $f$ such that $f(x^\top y) = \phi(x)^\top \phi(y)$, which can lack interpretability and compatibility with classic attention mechanisms.

However, one may consider nonlinear functions $f$ which do not admit a kernel; for example, the class of functions $f(x) = \log^c(|x| + 1)$ for $c > 0$ is used in Levy and Goldberg [2014], Li et al. [2015] to compute implicit word embeddings and corresponding generative models. Within this application domain, our work thus provides answers to the question: *for which classes of functions $f$ can one efficiently compute low-rank approximations $U^\star, V^\star$ such that $f(UV) \approx U^\star V^\star$, especially when $f(x^\top y)$ does not admit a kernel structure $\mathcal{K}(x, y)$?*

## 2 Preliminaries

We let our implicit matrix $A = f(UV)$ be $n \times d$ where $U \in \mathbb{R}^{n \times r}$ and $V \in \mathbb{R}^{r \times d}$ and $n \geq d$ without loss of generality, where $f : \mathbb{R} \to \mathbb{R}$ is a scalar function. We will also assume that $d = n^{\Omega(1)}$ and it is often the case in empirical settings that $d = n$. We say that a matrix $A$ is positive semi-definite (PSD) if it is symmetric and only has non-negative eigenvalues; a function $f$ is a kernel function if $f(UU^\top)$ is PSD for any matrix $U$. The $i$-th leverage score $\ell_i$ of matrix $B \in \mathbb{R}^{n \times d}$ is equal to its sensitivity, i.e. $\ell_i = \sup_{x \in \mathbb{R}^d} \frac{(B_i^T x)^2}{\|Bx\|_2^2} = \sup_{y \in \mathrm{colspan}(B)} \frac{y_i^2}{\|y\|_2^2}$, where $B_i$ is the $i$-th row of $B$. The Khatri-Rao product of $B \in \mathbb{R}^{n \times d}$ is $C \in \mathbb{R}^{n \times d^p}$, where the $i$-th row of $C$ is $B_i$ tensored with itself $p$ times. We use the standard notation that $\tilde{O}(f)$ is $O(f \, \mathrm{poly}(\log f))$.

For the approximate rank $k$ LRA problem, we want to find $U' \in \mathbb{R}^{n \times k}$ and $V' \in \mathbb{R}^{k \times n}$ such that

$$\|A - U'V'\| \leq (1 + \epsilon) \min_{\tilde{U} \in \mathbb{R}^{n \times k}, \tilde{V} \in \mathbb{R}^{k \times d}} \|A - \tilde{U}\tilde{V}\|$$

where $\|\cdot\|$ denotes the Frobenius norm, unless otherwise specified and we denote $[A]_k = \tilde{U}^\star \tilde{V}^\star$ as some rank $k$ matrix that minimizes the objective. Note this approximation is a relative error approximation guarantee, but can be analogously defined for additive error. For stronger lower bounds, we consider a weakened version of this problem given by outputting only a best rank $k$ projection, specifically we want to find orthogonal $W \in \mathbb{R}^{n \times k}$: $\|A - AWW^\top\| \leq (1+\epsilon)\|A - [A]_k\|$.

Our lower bounds will rely on reductions from the conditional hardness of OVP and Max-IP, whose hardness comes from SETH. We observe, from our remarks, that some more restrictive structure can be placed on the input vector sets $A, B$ to OVP without removing the hard instances.

**Assumption 1.** *(Hardness of Orthogonal Vectors Problem (OVP)[Williams, 2005]). Let $A = \{a_1, \ldots, a_n\}$ and $B = \{b_1, \ldots, b_d\}$ be sets, where $a_i, b_j \in \{0, 1\}^r$ are binary vectors for all $i \in [n] = \{1, 2, \ldots, n\}, j \in [d]$. Any algorithm which given any input $(A, B)$ decides with constant probability if there is at least one pair of vectors $a \in A$ and $b \in B$ such that $a^T b = 0$ requires $(nd)^{1-o(1)}$ time, provided $r = \omega(\log n)$ and $r = n^{o(1)}$. Observe that the lower bounds also hold when $A = B$ is enforced.*

**Remark 2.** *Previous work [Williams, 2005, Vassilevska Williams, 2015] has shown that Assumption 1 is true for $n = d$ unless the Strong Exponential Time Hypothesis (SETH, Impagliazzo and Paturi [2001]) is false. Given this assumption, one can handle general $n$ and $d$ by a padding argument: if one could solve OVP with an algorithm $\mathcal{A}$ running in at most $(nd)^{1-C}$ time for a constant $C > 0$, and without loss of generality $n \geq d$, then one could solve the problem when $|A| = |B| = n$ by splitting $B$ into $\Theta(n/d)$ disjoint sets each of size at most $d$, and solving the problem on each disjoint set in total time less than $(nd)^{1-C} \cdot \Theta(n/d) = n^{2-C}/d^C \leq n^{2-C}$, contradicting the assumption in the $|A| = |B| = n$ case.*

**Remark 3.** *In Assumption 1 we can enforce more restrictive structure on the input sets $A, B$ without making the problem easier. Specifically, we can assume that there are at most $n^{o(1)}$ distinct pairs with $a \in A$ and $b \in B$ for which $a^T b = 0$. Indeed, otherwise by sampling $(nd)^{2-\Omega(1)}$ pairs at random and checking if $a^T b = 0$, we would solve the OVP problem in $(nd)^{1-\Omega(1)} \cdot r = (nd)^{1-\Omega(1)}$ time, using that $r = n^{o(1)}$. This would contradict Assumption 1.*

**Assumption 4.** *(Hardness of Apx-Max-IP$_{n,d}$ Problem, Definition 2.1, Remark 2.2, and Lemma 4.1 of Chen and Williams [2019]). Two sets $A = \{a_1, \ldots, a_n\}$ and $B = \{b_1, \ldots, b_n\}$ are given, where $a_i, b_i \in \{0, 1\}^s$ are binary vectors for all $i \in [n]$, with $s = \omega(\log(n))$. Let $m = \max\limits_{a \in A, b \in B} a \cdot b$. Any algorithm which outputs a number $\tilde{m} \in [m/100, m]$ with constant probability requires $(nd)^{1-o(1)}$ time.*

**Remark 5.** *As in Remark 2 and Remark 3, we can reduce the general $n$ and $d$ case to the case $n = d$ of previous work [Chen and Williams, 2019], and we can also enforce that there are at most $n^{o(1)}$ pairs $a \in A$ and $b \in B$ for which $a \cdot b \geq m/100$, as otherwise sampling would solve the problem in less than $(nd)^{1-o(1)}$ time.*

## 3 LRA Runtime Lower Bounds

We show that the complexity of low rank approximation of an entrywise transformed matrix turns out to heavily depend on the type of entrywise transformation. Based on standard complexity assumptions, we show an $(nd)^{1-o(1)} = n^{1+\Omega(1)}$ time lower bound for entrywise function $f(x) = |x|^p$ for odd integers $p$ in Theorem 3.1 below, while we show a weaker $2^{\Omega(p)}$ lower bound for any integer in Theorem 3.3, which becomes $(nd)^{1-o(1)}$ for $p = \Theta(\log n)$. This is no accident, as we show a matching $2^{O(p)} n^{1+o(1)}$ upper bound in Theorem 4.1. Furthermore, our lower bounds extend to functions that are approximately odd-degree polynomials of $|x|$, even if the degree is small. Specifically, this includes the commonly used $f(x) = \log(|x| + 1) \approx |x|$.

Our proofs use variations of the Orthogonal Vectors Problem (OVP) to create two types of instances of implicit low rank approximation to create a fast $O(n^{1+o(1)})$ time algorithm to solve OVP (see Algorithm 1) when given access to a low rank approximation algorithm with constant relative error and small constant additive error guarantees. Let $A, B$ be the set of input vectors of OVP. Then, in our reduction, if there is a pair of input vectors which are orthogonal, then applying `LRA`, we can find either 1) a column span deviation of the low rank approximation or 2) a *sparse* vector in the column span of the low rank approximation. Note that we do not look at the exact column spans, but allow an additive error depending on the additive error of the output.

In the latter case, we note that our reduction structure ensures that all entries in this vector have the same magnitude. Consequently, since the column span of the output has low rank, each such entry in the support of this sparse vector corresponds to a large *leverage score*, and so to find the support of this unknown sparse column we can use algorithms to compute or approximate all leverage scores given the low rank factorization of the entrywise transformed matrix. This enables us to find a small superset of the support of the sparse vector, and then brute-force which pairs of entries in the superset correspond to vectors in the original OVP problem that intersect. The overall time in the reduction is negligible compared to the time to solve OVP, and therefore it must be that finding the factorization of the entrywise transformed matrix itself was expensive.

In the former case, it follows that the *column spans* of the output are necessarily far from each other in the two cases, and we can quickly check this and therefore solve the OVP problem. Together, it follows that the time for finding the factorization of the entrywise transformed matrix itself must have been large. While there are quantitative differences in the two cases of odd and even integers $p$, the proofs both follow this strategy.

---

**Algorithm 1** OVP to LRA Reduction

---

**Input:** Sets $\mathbf{A} = \{a_1, \ldots, a_n \in \{0,1\}^s\}$ and $\mathbf{B} = \{b_1, \ldots, b_n \in \{0,1\}^s\}$
**Output:** YES if there exists $a^\top b = 0$ and NO otherwise.
  1: Choose $c \in \{-1, 1\}^n$ uniformly at random
  2: Let $\mathbf{U} = [\mathbf{A} \mid c] \in \mathbb{R}^{n \times (s+1)}$ and $\mathbf{V}^\top = [\mathbf{B} \mid c] \in \mathbb{R}^{n \times (s+1)}$ and $r = s + 1$.
  3: Let $\mathbf{W} = \mathtt{LRA}(\mathbf{U}, \mathbf{V}, f = |x|^p)$ for constant relative error and $\alpha$ additive error
  4: Compute $\mathbf{U}'' = \mathbf{U} \otimes \ldots \otimes \mathbf{U} \in \mathbb{R}^{n \times r^p}$, $\mathbf{U}$ tensored with itself $p$ times and similarly for $\mathbf{V}''$.
  5: Let $r_i^2 = \|\mathbf{U}''\mathbf{V}''e_i\|_2^2 - \|\mathbf{U}''\mathbf{V}''\mathbf{W}\mathbf{W}^\top e_i\|_2^2$ for $i$-th column     ▷Calculate column distances of $\mathbf{U}''\mathbf{V}''$ to column span of $\mathbf{W}$.
  6: If any of $r_i^2 > 1.01\alpha$, output YES                     ▷$\alpha$ is any upper bound on additive error of $\mathtt{LRA}$
  7: Compute $1/2$-approximate leverage scores $\ell_i$ of $\mathbf{U}''$ and let $S = \{i \in [n] | \ell_i \geq 1/(100n^{o(1)})\}$
     ▷Finds the representative subset and the threshold is given by the OVP assumption.
  8: Compute all dot products between $a_i$ and $\mathbf{B}$ for all $i \in S$. Output YES if there exists a pair such that $a_i^\top b_j = 0$ and NO otherwise

---

## 3.1  General Functions of $|x|^p$ for odd integers $p$

In the following theorem we will also allow for a tiny amount of additive error, as this will be useful for our later lower bound applications where we approximate other functions using a Taylor series and reduce from the following theorem.

**Theorem 3.1.** *Suppose $n \in \mathbb{N}$, $f(x) = |x|^p$ for an odd integer constant $p = O(\log(n))$, and $r = O(\log(n))$. There is a positive integer $k = n^{o(1)}$, such that for any approximation factor $\Delta \geq 1$ and any constant $\alpha < 2$, any possibly randomized, algorithm which given any input $U \in \mathbb{R}^{n \times r}$ and $V \in \mathbb{R}^{r \times d}$ outputs $W \in \mathbb{R}^{n \times k}$ satisfying*

$$\|f(U \cdot V)WW^\top - f(U \cdot V)\|_F^2 \leq \Delta \cdot \|[f(U \cdot V)]_k - f(U \cdot V)\|_F^2 + \alpha,$$

*with constant probability, requires $(nd)^{1-o(1)}$ time, under Assumption 1. Further, this holds even if $U = V^T$. Here $[f(U \cdot V)]_k$ denotes the best rank-$k$ approximation to $f(U \cdot V)$ in the Frobenius norm.*

*Proof.* Suppose $A$ and $B$ are input sets to the OVP problem of Assumption 1 with parameter $r = s+1$, where $s$ is the dimension of the points. We let the rows of $U$ be the points in $A$, but we append one additional dimension, represented as column vector $c$ to $U$ that is chosen uniformly at random in $\{-1, 1\}^n$. Similarly, the columns of $V$ correspond to the points in $B$, but we append one additional row to $V$, which is equal to $c^\top$. Observe that if $A = B$, then necessarily $U = V^\top$.

Let Case 1 be when there is no $a \in A$ and $b \in B$ with $a^T b = 0$, and Case 2 be when there is an $a \in A$ and a $b \in B$ with $a^T b = 0$, and we are deciding whether we are in Case 2. We claim that our algorithm (Algorithm 1) will always output NO when we are in Case 1 and will output YES with constant probability when we are in Case 2. This clearly suffices to solve OVP with constant probability.

Notice that if there is no $a \in A$ and $b \in B$ for which $a^T b = 0$, then for all $a \in A$ and $b \in B$, we have $a^T b \geq 1$. Consequently with probability 1 over the choice of column vector $c$, for all $i \in [n]$ and $j \in [d]$, $U_i V_j \geq 0$, where $U_i$ is the $i$-th row of $U$ and $V_j$ is the $j$-th column of $V$. Consequently, $f(U_i V_j) = (U_i V_j)^p$. Thus, $f(U \cdot V) = U'' V''$, where each row of $U'' \in \mathbb{R}^{n \times r^p}$ is the Khatri-Rao product of itself $p$ times, and each column of $V'' \in \mathbb{R}^{r^p \times d}$ is the Khatri-Rao product of itself $p$ times.

Observe that the rank of $f(U \cdot V)$ is at most $\text{rank}(U'') + n^{o(1)} \leq (s+1)^p + n^{o(1)}$ since by Remark 3 we have at most $n^{o(1)}$ dot product pairs that are zero, implying that $f(U \cdot V)$ can be obtained from $U'' \cdot V''$ by changing at most $n^{o(1)}$ entries. It follows that there is a value $k \leq n^{o(1)} + (s+1)^p$ such that, in both cases, for any multiplicative approximation factor $\Delta \geq 1$, necessarily the output $W$ satisfies $\|f(U \cdot V)WW^\top - f(U \cdot V)\|_F^2 \leq \alpha$ with constant probability.

Next, we compare the column span of $W$ to that of $U'' \cdot V''$. In Case 1, there is no pairwise dot product that is zero, so we have that $f(U \cdot V) = U'' V''$. Therefore, each column of $U'' \cdot V''$ has squared distance at most $\alpha$ to the column span of $W$.

Note that we can compute all squared distances of $U'' V''$ to the column span of $W$ in $O(n^{1+o(1)})$ time. This is so that we can detect whether we are in case 1, where $U'' V''$ has at squared distance at most $\alpha$ to the column span of $W$, or we are in case 2. Specifically, we compute the squared distance by the Pythagorean theorem and since $W$ is orthogonal:

$$r_i^2 = \|U'' \cdot V'' e_i\|_2^2 - \|U'' V'' WW^\top e_i\|_2^2$$

and for each $i$, we can compute this in $n^{o(1)}$ time, given the matrices we have precomputed in the previous paragraph. Thus, we can compute all squared distances up to additive $1/\text{poly}(n)$ in $n^{1+o(1)}$ time. If we see that any squared distance is larger than $\alpha + 1/\text{poly}(n)$, we know we are in Case 2.

Therefore, we may assume in what follows that each column of $U'' \cdot V''$ has squared distance at most $\alpha + 1/\text{poly}(n)$ from the column span of $W$. In this case we can also determine in $n^{1+o(1)}$ time, with constant probability, whether we are in Case 2. To do so, we apply a *leverage score* approximation algorithm [Clarkson and Woodruff, 2013] to find a small representative subset $A' \subset A$ such that $|A'| = n^{o(1)}$ and if there exists $a \in A, b \in B$ such that $a^\top b = 0$, then $a \in A'$. Specifically, suppose there is an $a \in A$ and a $b \in B$ for which $a^T b = 0$, and suppose $U_i$ extends $a$ by one coordinate and $V_j$ extends $b$ by one coordinate. Then with probability at least $1/2$, $U_i^T V_j = -1$, and so $f(U_i^T V_j) = 1$. Let us condition on this event. Then we have the following claim in this case.

**Claim 6.** *Let $a \in A$ and $b \in B$ be such that $a^T b = 0$, and the corresponding row $U_i$ and column $V_j$ satisfy $U_i^\top V_j = -1$. Also, suppose each column of $U' \cdot V'$ has squared distance at most $\alpha + 1/\text{poly}(n)$ from the column span of $U''$. Then, in $n^{1+o(1)}$ time, we can find a representative subset $A' \subset A$ of size $n^{o(1)}$ such that $a \in A'$ with high probability.*

*Proof.* (of Claim) First note that by Remark 3, there can be at most $n^{o(1)}$ coordinates in the $j$-th column of $f(U \cdot V)$ which differ from their corresponding coordinate value in the $j$-th column of $U'' V''$, since each such difference corresponds to a pair of points $a \in A$ and $b \in B$ in the OVP problem for which $a^T b = 0$. Hence, since $f(U \cdot V)$ has squared distance at most $\alpha + 1/\text{poly}(n)$ from the column span of $W$, then consider the following vectors by taking the difference of the $j$-th column of $f(U \cdot V)$ and the $j$-th column of $U'' \cdot V''$, we would have vectors $v$ denoting the difference vector and $e$ being the residual vector of $v$ projected on the column span of $U''$, with the following properties:

- $v$ contains at most $n^{o(1)}$ values, each of value equal to 2,
- $\|e\| \leq \alpha + 1/\text{poly}(n)$
- $v + e$ is in the column span of $U''$.

Let $S \subset [n]$ be the set of of indices $i$ for which $\frac{(v+e)_i^2}{\|v+e\|_2^2} \geq 1/n^{o(1)}$. Since $\alpha + 1/\text{poly}(n)$ is at most a constant strictly less than 2, it follows that for each $i$ in the support of $v$, since $v_i = 2$, we have that $\frac{(v+e)_i^2}{\|v+e\|_2^2} \geq 1/n^{o(1)}$. Since also $\|v + e\|_2^2 \leq n^{o(1)}$, the support of $v$ is included in $S$ and $|S| \leq n^{o(1)}$.

Recall the definition of leverage scores $\ell_i$ from Section 2 and that the sum of the $n$ leverage scores of $B \in \mathbb{R}^{n \times t}$ is exactly equal to the rank of $B$, which is at most $t$. Since $v + e$ is in the column span of $U''$, the $i$-th leverage score of $U''$ satisfies $\ell_i(U'') \geq (v+e)_i^2 / \|v+e\|_2^2 = \Omega(1/n^{o(1)})$.

Consequently, each coordinate $i$ in $S$ satisfies $\ell_i(U'') = \Omega(1/n^{o(1)})$. It is known [Clarkson and Woodruff, 2013] how, given a matrix $B$, in $O(nt \log n) + t^{O(1)}$ time, one can compute a list $\ell'_1, \ldots, \ell'_n$ with $\ell'_i = \Theta(\ell_i)$ for all $i$ with probability $1 - 1/n^{100}$. Consequently, using that $\text{rank}(U'') \leq r^p$, given $U''$ one can find a superset $T$ containing $S$ for which $|T| = O(r^p n^{o(1)})$ in $n^{1+o(1)}$ time, where recall $r = s + 1$ and $s = n^{o(1)}$ and $p$ is constant. Note that our bound on $|T|$ follows since we just need to keep the leverage scores that are $\Omega(1/n^{o(1)})$ and the sum of all leverage scores is at most $r^p = n^{o(1)}$, recalling $r = s + 1$ and $s = n^{o(1)}$. $\qquad\square$

Continuing the proof of Theorem 3.1, by our claim, when we are in Case 2, with at least constant probability, we find in time $O(n^{1+o(1)})$ a representative subset $A'$ of size $O(n^{o(1)})$. Given our subset $A'$, one can then compute all pairs of dot products between the points in $A'$ and the points in $B$ in $O(n^{o(1)}ds)$ time, and since we may assume $n \geq d$ without loss of generality, we can compute all such pairs in $n^{1+o(1)}$ time. Lastly, if no such $a, b$ exist, when we are in Case 1, this algorithm cannot err. Therefore, we have an algorithm that can solve $OVP$, which then violates Assumption 1, using that $(nd)^{1-o(1)} = n^{1+\Omega(1)}$. $\qquad\square$

Our lower bound techniques also apply to a number of important function $f$ that do not have the form of $|x|^p$ for an integer $p$. We show that our lower bounds for LRA in fact holds for any function $g(x) = f(|x|)$, where $f(x)$ is a function that admits a Taylor expansion with a dominant term of $x^p$, for odd $p$, around 0. Of particular interest in natural language processing [Liang et al., 2020, Jiang et al., 2021] is the function $g(x) = \log(1 + |x|)$, where satisfies $g(x) = f(|x|)$, where $f(x) = x + O(|x|^2)$ and thus we can appeal to our lower bound with $p = 1$.

**Theorem 3.2.** *Suppose $n \in \mathbb{N}$, $r = O(\log(n))$, and $g(x) = f(|x|)$, where $f$ admits a Taylor expansion $f(x) = c_p x^p + O(|x|^{p+1})$ for odd $p = O(\log(n))$. There is a positive integer $k = n^{o(1)}$, such that for any approximation factor $\Delta \geq 1$, any, possibly randomized, algorithm which given inputs $U \in \mathbb{R}^{n \times r}$ and $V \in \mathbb{R}^{r \times d}$ outputs $W \in \mathbb{R}^{n \times k}$ satisfying*

$$\|g(U \cdot V)WW^\top - g(U \cdot V)\|_F^2 \leq \Delta \cdot \|[g(U \cdot V)]_k - g(U \cdot V)\|_F^2$$

*with constant probability, requires $(nd)^{1-o(1)}$ time, under Assumption 1. Further, this holds even if $U = V^\top$.*

### 3.2 Polynomials of All Integers $p$

We now give the proof for $p$-degree polynomials for all integers $p$, which is weaker than Theorem 3.1 for odd integers, but this is the first such lower bound for even integers. The weaker bound cannot be substantially improved since in this setting $f(x) = x^p$, so $f(UV)$ is in fact a kernel and matrix vector multiplication can be performed in $O(nr^p)$ time. For intuition why our lower bound techniques do not extend, recall that our previous reduction forces the absolute value operation to essentially alter entries of $(UV)^p$ but only at entries of the original with zero dot product. Therefore, we can write as a sum of a low rank matrix (from tensor product) and a sparse matrix, whose sparse entries now represent the OVP pairs. However, when $p$ is even, the absolute value operation leaves the entries unchanged and does not induce the additional sparse matrix. Therefore, our lower bounds for this setting rely on a slightly different variant of OVP, specifically quadratic lower bounds for finding the maximum dot product (Assumption 4).

Intuitively, our new reduction is as follows: Let OPT be the maximum inner product and by assumption, consider this small set of large inner product pairs, which represents a small number of entries in that are large in magnitude so that when you apply a threshold at OPT/100, the resulting matrix is sparse. Since $f(x) = x^p$ amplifies the magnitude differences, it follows that $(UV)^p$ is much closer relatively to an approximately sparse, and therefore low rank, matrix. Therefore an approximate low rank approximation (LRA) algorithm can recover this sparse low rank matrix well enough so that the span of the approximate matrix can be used, via leverage score computations, to solve the APX-Max-IP problem.

**Theorem 3.3.** *Let $U \in \mathbb{R}^{n \times r}$ and $V \in \mathbb{R}^{r \times d}$ be given with $r = O(\log(n))$. Suppose $f(x) = x^p$ for an integer $p \geq 1$. There is a positive integer $k = n^{o(1)}$, such that for any approximation factor $\Delta \geq 1$, any, possibly randomized, algorithm which outputs $W \in \mathbb{R}^{n \times k}$ satisfying $\|f(U \cdot V)WW^\top - f(U \cdot V)\|_F^2 \leq \Delta \cdot \|[f(U \cdot V)]_k - f(U \cdot V)\|_F^2$, with constant probability for any constant $\Delta > 1$, requires $\min((nd)^{1-o(1)}, 2^{\Omega(p)})$ time, under Assumption 4.*

The lower bound for general functions $f(x)$ in Theorem 3.2 with odd-degree dominating term in its Taylor expansion works because we can scale down the entries in our input matrix so that our relative-error approximation guarantee is still preserved but the function is largely approximated by only the leading polynomial term. By a combination of the same taylor expansion argument with a slight more general version of Theorem 3.3 that handles small additive error, we note that we can extend our results to when $p$ is even.

We note that our lower bounds do not generalize to the case when $U = V$ because as mentioned in our intuitive introduction, our reduction hinges on the sparse low-rank structure of $(UV)^p$. However, when $U = V$, the sparsity structure is broken as the diagonal of the matrix is now larger than the maximum inner product of two different vectors, and this destroys the low rank structure. In some sense, the diagonal of our matrix is forced to include the dot product of $u_i$ with itself and this shifts the entire matrix by a large multiple of the identity, crucially removing the sparse + low-rank structure that we exploited in our lower bound argument before. Indeed, as we remark below, the resulting positive semidefinite (PSD) structure allows us to derive a fast LRA algorithm and our lower bound no longer holds in this setting.

**Remark 7.** *Recall that when $U = V^\top$, there is an $nr(k/\epsilon)^{\omega-1}$ subquadratic time algorithm when $k = O(n^{o(1)})$, for $(1 + \epsilon)$-relative error approximation, but reducing the $U \neq V^\top$ case to the $U = V^\top$ case requires blowing up $k$ to $k + r^p$. Therefore, our lower bounds imply that the $U \neq V^\top$ case is strictly more difficult in terms of runtime in certain settings.*

## 4 LRA Algorithms from Matrix Vector Products

In this section, we can show upper bounds for low rank approximation of entrywise transformed products that are $O(n^{1+o(1)})$ when $f(x)$ represents a kernel matrix but $U \neq V^\top$. Specifically, we focus on polynomial functions of the form $f(x) = x^p$ and demonstrate that our lower bounds are tight for relative error LRA. Note that we have shown that low rank approximation guarantees for PSD matrices when $U = V^\top$ cannot translate to the case when $U \neq V^\top$. Still, we demonstrate that relative error low rank approximation is possible in $n^{1+o(1)}$ time for polynomial kernels with even degree, although there will be an exponential dependence on $p$ for relative error low rank approximation. Our relative error algorithms are relatively standard and rely on low rank projections by using matrix vector products to perform randomized sketching to reduce our row or column dimension to $O(\text{poly}(k/\epsilon))$ [Woodruff, 2014].

**Theorem 4.1.** *Let $U \in \mathbb{R}^{n \times r}$ and $V \in \mathbb{R}^{r \times d}$. Suppose $f(x) = x^p$ for an even integer $p \geq 1$ and $k < r^p$. For any approximation factor $\epsilon > 0$, there is an algorithm that outputs $U' \in \mathbb{R}^{n \times k}$ and $V' \in \mathbb{R}^{k \times d}$ satisfying $\|U' \cdot V' - f(U \cdot V)\|_F^2 \leq (1 + \epsilon) \cdot \|[f(U \cdot V)]_k - f(U \cdot V)\|_F^2$ with constant probability with runtime $O((n + d)r^p k/\epsilon^3 + \text{poly}(r^p/\epsilon))$.*

*Proof.* Note that $f(U_i V_j) = (U_i V_j)^p$. Thus, we may rewrite $f(U \cdot V) = U''V''$, where each row of $U'' \in \mathbb{R}^{n \times r^p}$ is the Khatri-Rao product of itself $p$ times, and each column of $V'' \in \mathbb{R}^{r^p \times d}$ is the Khatri-Rao product of itself $p$ times. Note that the rank of $U''V''$ is at most $d^p$.

Let $S$ be a random Gaussian sketching matrix with $O(k/\epsilon)$ rows, so we know that these matrices satisfy the $(\sqrt{\epsilon/k}, 9/10, l)$-JL property [Woodruff, 2014]. Furthermore, let $R$ be a random Gaussian matrix with $O(\min(k/\epsilon^3, r^p/\epsilon^2))$ columns, so we know that it is a $(1+O(\epsilon))$ $\ell_2$ subspace embedding of the row space of $SU''$. Then, by Theorem 47 of Woodruff [2014], the following is true with constant probability

$$\|(U''V''R)(SU''V''R)^+(SU''V'') - f(UV)\|_F^2 \leq (1 + \epsilon) \cdot \|[f(U \cdot V)]_k - f(U \cdot V)\|_F^2$$

Finally we bound the runtime of computing this product. Note that we may compute $SU''$ and $V''R$ in $nr^p \cdot (k/\epsilon + \min(k/\epsilon^3, r^p/\epsilon^2))$ time. Then, note that the remaining products can be computed in

poly$(kr^p/\epsilon) = $ poly$(r^p/\epsilon)$ time. And lastly, the rank $k$ approximation follows from solving for the low rank approximation on the restricted subspace $SU''V''R$ of rank and letting $Z = [U''V''RU]_k$, where $U$ is the orthonormal basis such that $UU^\top$ is the projection onto the row space of SAR. $\qquad\square$

We also provide an additive error guarantee on the low rank approximation guarantees that depends polynomially on $p$, as opposed to the tight exponential dependence in the relative error setting. This implies that additive error LRA is strictly easier and can be accomplished in subquadratic time when $p = \Omega(\log(n))$. Note that lower bounds for additive error are not emphasized in this paper as the upper bound is already quite competitive and simply outputting requires $\Omega(n * \text{poly}(p))$.

Our guarantees follow from tensor-based sketching techniques that are applied along each tensor dimension to remove the exponential dependence on $p$. We note that this results in an additive error term that is on the order of the $p$-norms of $U, V$. This term is related to the absolute error guarantees of [Han et al., 2020], which also depends on the product of the $p$-norms of Euclidean norms of the rows of $U, V$.

---

**Algorithm 2** TensorSketch LRA

---

**Input:** Matrices $\mathbf{U} \in \mathbb{R}^{n \times r}, \mathbf{V} \in \mathbb{R}^{r \times d}$, $p > 0$ even integer
**Output:** rank $k$ approximation of $f(UV)$, where $f(x) = |x|^p$
  1: Let $\mathbf{T}$ be a TensorSketch with $m = O(p\epsilon^{-2})$ rows $\qquad\qquad$ ▷See Ahle et al. [2020]
  2: Compute $\mathbf{U}''' = \mathbf{U}''\mathbf{T}^\top$ and $\mathbf{V}''' = \mathbf{T}\mathbf{V}''$, where $\mathbf{U}'' \in \mathbb{R}^{n \times r^p}, \mathbf{V}'' \in \mathbb{R}^{r^p \times d}$ are $\mathbf{U}, \mathbf{V}$ tensored themselves $p$ times.
  3: Let $\mathbf{S}, \mathbf{R}$ be random Gaussian matrices with $O(k/\epsilon)$ rows and $O(p/\epsilon^4)$ columns respectively
  4: Compute an orthonormal basis $\mathbf{P}$ of row span of $\mathbf{S}\mathbf{U}'''\mathbf{V}'''\mathbf{R}$. $\qquad\qquad$ ▷This implies $\mathbf{P}\mathbf{P}^\top = (\mathbf{S}\mathbf{U}'''\mathbf{V}'''\mathbf{R})^+(\mathbf{S}\mathbf{U}'''\mathbf{V}'''\mathbf{R})$
  5: Compute the rank $k$ decomposition: $\mathbf{U}'\mathbf{V}' = [\mathbf{U}'''\mathbf{V}'''\mathbf{R}\mathbf{P}]_k$. $\qquad$ ▷$[\mathbf{M}]_k$ is the best rank $k$ approximation to $\mathbf{M}$
  6: Output $\mathbf{U}', \mathbf{V}'\mathbf{P}^\top(\mathbf{S}\mathbf{U}'''\mathbf{V}'''\mathbf{R})^+\mathbf{S}\mathbf{U}'''\mathbf{V}'''$

---

**Theorem 4.2.** *Let $U \in \mathbb{R}^{n \times r}$ and $V \in \mathbb{R}^{r \times d}$. Suppose $f(x) = x^p$ for an even integer $p \geq 1$ and $k < r^p$. For any approximation factor $\epsilon > 0$, there is an algorithm (Algorithm 2) that outputs $U' \in \mathbb{R}^{n \times k}$ and $V' \in \mathbb{R}^{k \times d}$ satisfying $\|U' \cdot V' - f(U \cdot V)\|_F^2 \leq (1 + \epsilon) \cdot \|[f(U \cdot V)]_k - f(U \cdot V)\|_F^2 + \epsilon^2 L^2$ with constant probability with runtime $O((n + d) \cdot \text{poly}(p, r, k, 1/\epsilon))$, where the additive term is given by $L^2 = (\sum_{i=1}^n \|U_i\|_2^{2p})(\sum_{i=1}^d \|V_i\|_2^{2p}) = \|U\|_{2p,2}^{2p}\|V^\top\|_{2p,2}^{2p}$.*

### 4.1 Lower Bounds on Matrix Multiplication

Our low rank approximation algorithms use a matrix vector multiplication subroutine $f(UV)z_i$ for $O(\text{poly}(k/\epsilon))$ different vectors $z_i$ as their main dimensionality reduction technique for subquadratic time upper bounds. This can be used to directly translate lower bounds for low rank approximation to implicit matrix vector multiplication for a wide range of scalar functions that extends previous work beyond the exponential function. We emphasize that these matrix-vector product lower bounds also imply that our LRA lower bounds are non-trivial and significantly strengthen those provided by previous works.

**Theorem 4.3** (LRA reduces to Matrix Vector Products)**.** *Let $U, V, f$ be as in Theorem 3.2. Then, any possibly randomized algorithm that for any vector $z \in \mathbb{R}^n$, outputs $f(UV)z$ up to $1/poly(n)$ entrywise error, with constant probability, requires $\Omega((nd)^{1-o(1)})$ time, under Assumption 1. This holds even when $U = V^\top$.*

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

## A Missing Lower Bound Proofs

*Proof of Theorem 3.2.* Let $U, V^T \in \{-1, 0, 1\}^{n \times r}$ be an instance of entrywise transformed low rank approximation with $g$. WLOG, by scaling our solution, we can let $c_p = 1$. Let $B = \text{poly}(n)$ be sufficiently large, and let $\tilde{U} = U/\sqrt{B}$ and $\tilde{V} = V/\sqrt{B}$. For each $i, j \in [n]$, for $B$ sufficiently large and using that $r = n^{o(1)}$,

$$
\begin{aligned}
g(\tilde{U}_i \tilde{V}_j) &= f(|\tilde{U}_i \tilde{V}_j|) = f(|U_i V_j|/B) = \frac{|U_i V_j|^p}{B^p} - O\left(\frac{|U_i V_j|^{p+1}}{B^{p+1}}\right) \\
&= \frac{|U_i V_j|^p}{B^p} - O\left(\frac{r^p}{B^{p+1}}\right) = h(\tilde{U}_i \tilde{V}_j) - O\left(\frac{r^p}{B^{p+1}}\right).
\end{aligned}
$$

where $h(x) = |x|^p$. Hence, if $\tilde{U}', \tilde{V}'$ are such that

$$
\|\tilde{U}' \cdot \tilde{V}' - g(\tilde{U} \cdot \tilde{V})\|_F^2 \leq \Delta \|[g(\tilde{U} \cdot \tilde{V})]_k - g(\tilde{U} \cdot \tilde{V})\|_F^2,
$$

then by the triangle inequality,

$$
\begin{aligned}
\|\tilde{U}' \cdot \tilde{V}' - h(\tilde{U} \cdot \tilde{V})\|_F &\leq \|g(\tilde{U} \cdot \tilde{V}) - h(\tilde{U} \cdot \tilde{V})\|_F + \sqrt{\Delta} \|[g(\tilde{U} \cdot \tilde{V})]_k - g(\tilde{U} \cdot \tilde{V})\|_F \\
&\leq O\left(\frac{nr^p}{B^{p+1}}\right) + \sqrt{\Delta} \|[g(\tilde{U} \cdot \tilde{V})]_k - g(\tilde{U} \cdot \tilde{V})\|_F.
\end{aligned}
$$

Setting $U' = \sqrt{B} \cdot \tilde{U}'$ and $V' = \sqrt{B} \cdot \tilde{V}'$ and scaling both sides by $B$, we have

$$
\begin{aligned}
\|U' \cdot V' - h(U \cdot V)\|_F &\leq O\left(\frac{nr^p}{B^p}\right) + B\sqrt{\Delta} \|[g(\tilde{U} \cdot \tilde{V})]_k - g(\tilde{U} \cdot \tilde{V})\|_F \\
&\leq O\left(\frac{nr^p}{B^p}\right) + B\sqrt{\Delta} \|[h(\tilde{U} \cdot \tilde{V})]_k - g(\tilde{U} \cdot \tilde{V})\|_F \\
&\leq O\left(\frac{nr^p}{B^p}\right) + B\sqrt{\Delta} \|[h(\tilde{U} \cdot \tilde{V})]_k - h(\tilde{U} \cdot \tilde{V})\|_F \\
&\quad + B\sqrt{\Delta} \|h(\tilde{U} \cdot \tilde{V}) - g(\tilde{U} \cdot \tilde{V})\|_F \\
&= O\left(\frac{\sqrt{\Delta} nr^p}{B^p}\right) + B\sqrt{\Delta} \|[h(\tilde{U} \cdot \tilde{V})]_k - h(\tilde{U} \cdot \tilde{V})\|_F \\
&= O\left(\frac{\sqrt{\Delta} nr^p}{B^p}\right) + \sqrt{\Delta} \|[h(U \cdot V)]_k - h(U \cdot V)\|_F,
\end{aligned}
$$

where in the second inequality we used that $[h(\tilde{U} \cdot \tilde{V})]_k$ has rank $k$ whereas $[g(\tilde{U} \cdot \tilde{V})]_k$ is the best rank-$k$ approximation to $g(\tilde{U} \cdot \tilde{V})$ in Frobenius norm. Note that the third line is triangle inequality and the fourth line follows our entrywise approximation bounds.

Now by AM-GM, using that $a \leq b + c$ implies that $a^2 \leq 2b^2 + 2c^2$ for $a, b, c \geq 0$, we have

$$
\|U' \cdot V' - h(U \cdot V)\|_F^2 = O\left(\frac{\Delta n^2 r^{2p}}{B^{2p}}\right) + 2\Delta \|[h(U \cdot V)]_k - h(U \cdot V)\|_F^2.
$$

Thus, $U' \cdot V'$ is a rank-$k$ approximation to $h(U \cdot V)$ with additive error $\frac{1}{\text{poly}(n)}$, by setting $B$ to be large enough, and relative error $2\Delta$. The total time to find $U'$ and $V'$ is the same as the time to solve entrywise transformed low rank approximation with respect to the function $g$ on inputs $\tilde{U}$ and $\tilde{V}$, and thus by Theorem 3.1 is at least $(nd)^{1-o(1)}$. Note that applying the case $U = V^\top$ in Theorem 3.1 establishes this theorem when $\tilde{U} = \tilde{V}^\top$. $\qquad\square$

*Proof of Theorem 3.3.* Suppose $A$ and $B$ are the input sets to the Apx-Max-IP$_{n,d}$ Problem of Assumption 4 with parameter $s = r$. We let the rows of $U$ be the points in $A$ and we let the columns of $V$ be the points in $B$. As in the proof of Theorem 3.1, we have $f(U_i^T V_j) = (U_i^T V_j)^p$ so $f(U \cdot V) = U'' \cdot V''$. By Remark 5, there are at most $n^{o(1)}$ entries of $f(U \cdot V)$ that are at least $(m/100)^p$.

Hence, if we set $k$ to be an appropriate value in $n^{o(1)}$, then

$$\|[f(U \cdot V)]_k - f(U \cdot V)\|_F^2 \leq (nd)(m/100)^{2p},$$

since one possible rank-$k$ approximation is just to zero out the at most $n^{o(1)}$ entries of $f(U \cdot V)$ that are at least $(m/100)^p$. By the correctness guarantee, with constant probability,

$$\|f(U \cdot V)WW^\top - f(U \cdot V)\|_F^2 \leq \Delta \cdot (nd)(m/100)^{2p} = O(nd)(m/100)^{2p}. \qquad \text{(A.1)}$$

We may assume WLOG that $p \geq C \log n$ for the moment, for a sufficiently large constant $C > 0$ since a trivial lower bound for outputting an LRA is $\Omega(n)$, so the inclusion of the $2^{\Omega(p)}$ term in the lower bound allows us to make this assumption. Therefore, we conclude that $f(U \cdot V)$ is in the column span of $W$ up to $1/\text{poly}(n)$ error. Consequently, consider the maximum entry $m$, which we can assume is at least 1, as otherwise all points in $A$ would have disjoint support from those in $B$ but this can be verified in $O(nr) = n^{1+o(1)}$ time, contradicting the $n^{1+\Omega(1)}$ lower bound for $d = n^{\Omega(1)}$ in Assumption 4.

It follows that if the maximum $m$ occurs in the $(i, j)$-th entry of $f(U \cdot V) = U''V''$, then there exists a vector in the column span of $U''$ for which the $i$-th entry is $[m^p(1 - \text{poly}(n)), m^p(1 + \text{poly}(n))]$ and all other entries are in the range $[-m^p/\text{poly}(n), m^p/\text{poly}(n)]$ since $(1/100)^p = 1/\text{poly}(n)$. Consequently, since the column span of $W$ is close to the column span of $U''$ up to $1/\text{poly}(n)$ error, we see that the column span of $W$ also contains these almost-sparse vectors that have $n^{o(1)}$ large entries. As argued before similarly in Theorem 3.1, the rows of these large dot products forces the row leverage scores of $W$ to be $\Omega(1)$. Since $W$ has rank $k = n^{o(1)}$, in $n^{1+o(1)}$ time we can find the set $S$ of $n^{o(1)}$ row leverage scores of $W$ that are $\Omega(1)$. We can then explicitly compute all dot products between pairs of vectors in $A \cap S$ and $B$ in $n^{1+o(1)}$ total time, at which point we can output the maximum dot product, which includes $m$. Therefore by Assumption 4, the total time to find $W$ is at least $(nd)^{1-o(1)}$.

$\square$

*Proof of Theorem 4.3.* This proof follows by reducing low rank approximation to matrix multiplication by the same algorithm as in Theorem 4.1. Specifically, let us set $k = n^{o(1)}$ and $n = d$ and note that our low rank approximation algorithm's runtime is dominated by computing $S \cdot f(UV)$ and $f(UV) \cdot R$, where $S, R$ are matrices with $\text{poly}(k/\epsilon)$ rows and columns, respectively. In fact, those operations are the only terms that incur a polynomial dependence on $n$.

Therefore, suppose there exists such a matrix multiplication algorithm that takes time $O(n^{2-c})$ for some $c$. Then this would directly imply that computing both $S \cdot f(UV), f(UV) \cdot R$ takes $O(n^{2-c})$ time as $[Sf(UV)]^\top = f(V^\top U^\top)S^\top$. By our guarantees in Theorem 47 of Woodruff [2014], this implies a constant relative error LRA in time $O(n^{2-c})$, which contradicts Theorem 3.2. $\square$

# B Missing Upper Bound Proofs

*Proof of Theorem 4.2.* Again, $f(U_i V_j) = (U_i V_j)^p$ so we may rewrite $f(U \cdot V) = U''V''$, where each row of $U'' \in \mathbb{R}^{n \times r^p}$ is the Khatri-Rao product of itself $p$ times, and each column of $V'' \in \mathbb{R}^{r^p \times d}$ is the Khatri-Rao product of itself $p$ times.

Using Theorem 1 of Ahle et al. [2020], we see that with probability 0.9, our approximate matrix product guarantees hold for $U'', V''$ such that if $\Pi$ is an $m \times r^p$ TensorSRHT matrix with $m = \Theta(p/\epsilon^2)$, then

$$\|U''\Pi^\top \Pi V'' - U''V''\|_F \leq \epsilon \|U''\|_F \|V''\|_F$$

Therefore, we can use the approximate low rank sketches again to approximately solve:

$$\min_{UV} \|U''\Pi^\top \Pi V'' - UV\|_F^2$$

Specifically, let $S$ be a random Gaussian sketching matrix with $O(k/\epsilon)$ rows. We know that these matrices satisfy the $(\sqrt{\epsilon/k}, 9/10, l)$-JL property Woodruff [2014]. Furthermore, let $R$ be a random

Gaussian matrix with $O(m/\epsilon^2)$ columns, so we know that it is a $(1 + O(\epsilon))$ $\ell_2$ subspace embedding of the row space of $SU''\Pi^\top$, which has rank at most $m$. Let $U''' = U''\Pi^\top$ and let $\Pi V'' = V'''$. Then, by Theorem 47 of Woodruff [2014], the following is true with constant probability

$$\|(U'''V'''R)(SU'''V'''R)^+(SU'''V''') - U'''V'''\|_F \leq (1 + \epsilon) \cdot \min_{U,V} \|UV - U'''V'''\|_F$$

Finally we bound the runtime of computing this product. Note that we may compute $U'''$ and $V'''$ in time $O(np(m + r))$ time. Then, computing $SU''', V'''R$ can be done in $nm \cdot (k/\epsilon + m/\epsilon^2)$ time. Lastly, the remaining products can be computed in $\text{poly}(kp/\epsilon)$ and the final rank $k$ decomposition can be computed and we can find $U', V'$ such that

$$\|U'V' - U'''V'''\|_F \leq (1 + \epsilon) \min_{U,V} \|UV - U'''V'''\|_F$$

Now, let $U^\star, V^\star$ be the optimal rank $k$ decomposition of the original problem of $f(UV)$, then by the guarantees of approximate matrix product and the triangle inequality, $\|U^\star V^\star - U''V''\|_F \geq \|U^\star V^\star - U''\Pi^\top \Pi V''\|_F - \epsilon\|U''\|_F\|V''\|_F$.

Therefore, we conclude that

$$\|U'V' - U''\Pi^\top \Pi V''\|_F \leq (1 + \epsilon)\|U^\star V^\star - U''\Pi^\top \Pi V''\|_F$$
$$\leq (1 + \epsilon)\|U^\star V^\star - U''V''\|_F + 2\epsilon\|U''\|_F\|V''\|_F$$

We end by rewriting $\|U''\|_F^2 = \text{Tr}(U''U''^\top) = \sum_{i=1}^n (U_i^\top U_i)^p = \sum_i \|U_i\|_2^{2p}$ and similarly for $\|V''\|_F^2$ and then applying AM-GM and noting that $(1 + \epsilon)^2 = 1 + O(\epsilon)$. $\qquad\square$

