# OpenReview forum: "Hardness of Low Rank Approximation of Entrywise Transformed Matrix Products"
_NeurIPS.cc/2023/Conference — NeurIPS 2023 poster_

### Official Review · Reviewer_26kb · 2023-06-28

**Soundness:** 3 good
**Presentation:** 3 good
**Contribution:** 3 good
**Rating:** 7
**Confidence:** 4

**Summary:**

This paper studies the problem of computing a low-rank approximation (LRA) for $f(UV)$, where $U\in \mathbb{R}^{n\times r}$, $V\in \mathbb{R}^{r\times n}$, and $f:\mathbb{R}\rightarrow \mathbb{R}$ applies to each entry of $UV$. This problem has very important applications in deep learning, in particular for Transformers and natural language processing. The main results of this paper consist of two parts:

For lower bounds, they show that we cannot get subquadratic relative error LRA generally when either 1) $U\ne V^\top$ even for PSD kernel functions or 2) for $f$ that are approximately polynomials of $|x|$ even for a constant degree.

For upper bounds, they show that for $f(x) = x^p$, there is an $O(n\cdot poly(r^p, k, 1/\epsilon))$-time algorithm for relative error LRA and $O(n \cdot poly(p, k, 1/\epsilon))$-time for additive error LRA.

Furthermore, their techniques can be generalized to obtain lower bounds for the implicit matrix-vector multiplications of the form $f(UV)z$.


**Strengths:**

This paper significantly advances our understanding of LRA with very solid results. Prior to this work, most of the research focuses on the symmetric setting where $U=V^\top$. This paper considers the more general, asymmetric setting. For relative error LRA and $f(x)=x^p$, their lower bound almost matches their upper bound in terms of the dependence on $p$. On the other hand, they also make progress in the symmetric setting. Their lower bound in the symmetric setting applies to a larger family of the function $f$ compared to previous results. Technically, they provide clean and novel reductions from the orthogonal vector problem (OV) to LRA via the fast computation of leverage scores. The reductions in this paper will be useful in proving new lower bounds for other numerical linear algebra problems. Additionally, this paper provides a clear review of many related works about LRA.



**Weaknesses:**

The algorithms proposed in this paper is quite straightforward given prior results  [Woodruff, 2014] and [Ahle et al., 2020]. Also, the lower bound for LRA with an even $p$ is weak. And the lower bounds do not depend on $r$ and $k$, the dimensions of the input matrices and output matrices, respectively. Moreover, there are some typos in the proofs that affects reading this paper.


**Questions:**

1. Line 157: define $B_i$.

2. Do the lower bounds for OV (Assumption 1) and Max-IP (Assumption 4) hold for randomized algorithms? They may be based on (randomized) SETH.

3. Line 262: $s\leq r^p$.$s$ has already been used to represent the number of vectors in OV. I guess the $s$ here is different?

4. Line 265: the derivation of the below equation via the Pythagorean theorem is unclear to me. And how does it imply that when we are in case 2, there is a column of $U’V’$ that is $\alpha+1/poly(n)$ far from the column span of $U’’$?

5. Line 287: how do you construct the vectors $v$ and $e$?

6. Line 306: where does the $n^{o(1)}dr^ps$ come from? Shouldn’t it be $n^{o(1)}ds$?

---

> ### Author Rebuttal · Authors · 2023-08-08
>
> Thank you for your positive review and careful reading of our paper.
>
> We address your questions below:
>
> 1. $B_i$ is the $i$-th row of matrix $B$.
>
> 2. Yes, the lower bounds do hold for randomized algorithms (and OVP is stated with constant probability). In general, by Yao’s minimax principle, deterministic lower bounds translate into lower bounds for randomized algorithms.
>
> 3. Generally, $n$ is used to represent the number of vectors in $OV$, and $s$ is defined in Theorem 3.1/Algorithm 1 and is used to denote the dimension of the input vectors. We will clear up any confusing notation.
>
> 4. Yes, we will clarify the notation and derivations. Generally, we are claiming that we can compute all squared distances of $U’V’$ to the column span of $U’'$ in $O(n^{1+o(1)})$ time. This is so that we can detect whether we are in case 1, where $U’V’$ has squared distance at most alpha to the column span of $U’’$, or we are in case 2 (the opposite case). To compute this, we can first compute orthogonal bases $Z$, $W$, respectively for $U’’$ and $U’$, and then we can compute the squared distance using the Pythagorean theorem, since the squared distance is equal to (total squared distance) - (projected squared distance).
>
> 5. Let $v$ = distance of $j$-th column of $U’’V’$’ to $j$-th column of $U’V’$, and $e$ = residual vector of $v$ projected on the column span of U’’, such that $v + e$ is in the span.
>
> 6. Yes, thanks that is indeed a typo from a previous version and it is fixed.

---

> > ### Comment · Reviewer_26kb · 2023-08-14
> >
> > I thank the authors for their response and clarification. I keep my score.

---

### Official Review · Reviewer_ExZq · 2023-07-06

**Soundness:** 2 fair
**Presentation:** 2 fair
**Contribution:** 3 good
**Rating:** 6
**Confidence:** 2

**Summary:**

The authors discuss a setting where one observes a scalar transformation of an r-rank matrix, f(UV) and wants to find the best k-rank approximation of it.

It is known that under various assumptions, if f(UU^t) is PSD then the problem is solvable in nearly linear time. It is also known that for various kernels f (eg Gaussian kernel), one needs quadratic time.

The authors discuss the cases where f are simple monomials, f(x)=x^p. They show a min(2^p,n^2) time lower bound based on SETH.  They also show a matching upper bound, among other results.




**Strengths:**

The paper seems novel and interesting. The connection with SETH is non-trivial and intruiguing.

**Weaknesses:**

I am failing to see the importance of choosing the x^p activation function (besides the log(1+|x|) example). Could the authors elaborate more on it?

**Questions:**

See above.

**Limitations:**

See above.

---

> ### Author Rebuttal · Authors · 2023-08-08
>
> Thank you for your positive comments.
>
> We address your comment regarding the “importance of choosing the $x^p$ activation function”: Note that we extend our base case results over $x^p$ to a broader class of functions (via a Taylor expansion) in Theorem 3.2. Specifically, our results also hold for any function $g(x) = f(|x|)$ where $f(x)$ is a function that admits a Taylor expansion with a dominant term of $x^p$ for odd $p$, around $0$. Examples include most activation functions, as they have a dominant linear term, such as $\text{sigmoid}(x)$ or unnormalized softmax $\exp(x)$, which behave like $1 + x$ around $0$.

---

> > ### Comment · Reviewer_ExZq · 2023-08-14
> >
> > I thank the authors for their response. Regarding the Taylor series argument, do you mean "f admits a Taylor expansion" for the whole real line or literally in a neighborhood "around 0"? This is important to clarify.
> >
> > I keep my score (6), with low confidence.

---

> > > ### Author Response · Authors · 2023-08-15
> > > **Re: Comment**
> > >
> > > When we say "$f$ admits a Taylor expansion", we are adopting the classic convention that $f$ can be represented by an infinite Taylor expansion (centered at 0) for the whole real line. However, in our proofs, we only use the approximation guarantees in a neighborhood around 0. However, for ease of reading, we generally assume the stronger condition.

---

### Official Review · Reviewer_GCRp · 2023-07-07

**Soundness:** 3 good
**Presentation:** 3 good
**Contribution:** 2 fair
**Rating:** 6
**Confidence:** 3

**Summary:**

This paper studies the problem of low rank approximation of the form f(U.V) where the matrices U,V $\in \mathbb{R}^{n \times r}$ and $r=O(\log n)$ and $f(.)$ is applied entrywise. The main contributions of the paper are the following:

1) The paper presents a $\Omega(n^{2-o(1)})$ time lower bound for getting relative error rank $k$ approximation to $f(U.V)$ for $k=n^{o(1)}$ and $f(x)$ admits a Taylor series expansion of the form $f(x)=x^p+O(|x|^{p+1})$ when $p$ is odd and $p=O(\log n)$. This lower bound holds for any $U, V$. Another $\min(n^{2-o(1)},2^{\Omega(p)})$ time lower bound, which only holds when $U \neq V^T$, is presented for the case when $f(x)=x^p$ for any general integer $p \geq 1$. The first lower bound proof for odd $p$ follows by a reduction of the Orthogonal Vectors Problem to the Low-Rank Approximation (LRA) problem of $f(U.V)$. The proof then proceeds by lower bounding the leverage score of rows corresponding to non-zero entries of a flat sparse vector in the column span of the low-rank approximation of $f(U.V)$. The second lower bound proof for a general $p$ follows by a reduction of the Approximate Maximum Inner Product problem respectively to the LRA problem.

2) The paper presents a $O(n . poly(2^p,k,\frac{1}{\epsilon}))$ time algorithm for getting a $1+\epsilon$ relative error low-rank approximation to $f(U.V)$ mathcing the lower bound. The algorithm uses random Gaussian sketching of the Khatri-Rao products of U and V followed by a low-rank approximation of the smaller sketches. The paper also presents a $O(n, poly(k,p, \frac{1}{\epsilon}))$ time algorithm for getting a low-ran approximation with additive $\epsilon . L$ guarantee where $L$ depends on the $p$-norms of $U,V$. Thi algorithm proceeds by doing a tensor sketch of the Kahtri-Rao products of $U$ and $V$.

3) Finally, the paper presents a lower bound of $\Omega(n^{2-o(1)})$ on matrix vector products of the form $f(UV)z$ for a vector $z \in \mathbb{R}^n$ using the fact that the algorithms use a matrix vector product subroutine.

**Strengths:**

1) Previous subquadratic time algorithms for relative error low-rank approximation for $f(U.V)$ assume that $U=V^T$ and $f(x)$ is a PSD kernel function. The quadratic lower bounds for relative error low-rank approximation of $f(U.V)$ for polynomial $f(.)$ show that both these assumptions are necessary for a subquadratic algorithm. This is an interesting result of broader interest to the sketching community. The lower bound proof techniques via reductions from OVP and MIP problems and bounding the leverage score of flat sparse vectors are also novel.

2) The paper is generally easy to follow for the most part.

**Weaknesses:**

1) Apart from the lower bounds of Theorem 3.1 and 3.3, the other results seem to be straightforward extensions of these results or, for the case of the upper bounds, applications of existing algorithms and results in the literature.

2) Though the results are interesting, the intuition behind some of the theoretical results presented in the paper are unclear.

     a) Specifically, it might be useful to explain more in detail why is there a separation in the lower bound between the $U=V^T$ and $U \neq V^T$ cases for general $p$.

     b) It might be good to explain why the lower bound for general $p$ extend to more general functions with Taylor series of the form $f(x)=x^p+O()$ like in the case for odd $p$. Where is the bottleneck for general $p$?

2) Though this is primarily a theoretical paper, in the context of Neurips, some experimental evaluation of the algorithms for LRA might be useful for understanding how well the algorithms perform well in practice.

There are also some typos and confusing notations in the paper:
1) In the line 260, $Z$ has dimension $n \times s$. The $s$ here is the rank of $U''$ and is different from the dimension of the  vectors $a_i$ and $b_i$.
2) In the statement of Theorem 4.1, on the RHS, it seems the $U,V$ in argument $\min_{U,V}$ should be changed to different notation as this is different from the input to the problem.

**Questions:**

Other than the points raised above, there doesn't seem to be any discussion on the lower bound for additive error so it is unclear how good the upper bound presented in Theorem 4.2 is. It might be useful to add a short discussion there.


**Limitations:**

I don't think there are any other limitations other than the ones mentioned above.

---

> ### Author Rebuttal · Authors · 2023-08-08
>
> Thank you for your careful review of our paper - the following are our responses to the main concerns raised:
>
> Addressing Weaknesses:
>
> 1. We emphasize that we presented our results in a way that highlights the novelty and difficulty of our lower bounds, which could be of general interest even in algorithm design. Our upper bounds, though novel to our knowledge, are indeed relatively mathematically straightforward compared to our lower bounds. However, their main purpose is to supplement the lower bounds, not necessarily in novelty or difficulty, but in showing that our derived lower bounds are in fact optimal. Therefore, including such upper bounds is important.
>
> 2. Yes, we will add more intuition behind the theoretical results, which we mention below:
>
>   * There is a separation of algorithmic time complexity for relative low rank approximation (LRA) between (1) the $U = V^T$ setting and (2) the $U \neq V^T$ setting. For the sake of illustration, let’s consider the case when $f(x) = x^{2p}$ for $p = \Omega(\log(n))$ sufficiently large and $k = O(1)$. In the setting 1, we have an $n * \text{poly}(\log n)$ time algorithm for constant factor LRA using the cited paper by Bakshi et al.,  2020. However, in setting 2, we provide a lower bound of nearly $n^2$. Therefore, we conclude that there is an inherent complexity separation in the two settings.
>
>   * The extension of the lower bound for general $p$ occurs because the Taylor expansion of the function is dominated by the $x^p$ term around $0$. Therefore, we can scale down the entries in our input matrix so that our relative-error approximation guarantee is still preserved but the function is dominated by the leading polynomial term (please see Theorem 3.2 for more details). In fact, this technique also generalizes to the general $p$ setting; however, it was not included for brevity and we will include this in the final version.
>
> 3. Experimental evaluation: while there are many papers focusing on empirical evaluations, our focus is on our lower bounds as they require novel arguments involving leverage scores, and stand out in a field that is dominated by positive results.
>
> Addressing the additional questions:
>
> * “No discussion on lower bounds for additive error” - Lower bounds for additive error LRA are not emphasized because the upper bound is already quite small and is based on the optimality of TensorSketch in Ahle et al., 2020. Specifically note that when $k = \text{poly}(p)$, simply outputting $U, V$ already takes $\Omega(n * \text{poly}(p))$ times and our upper bound on additive LRA is $O(n * \text{poly}(p))$. Our upper bounds for additive error were used to highlight a separation between additive and multiplicative error. We will add this discussion in our final version.
>
> We will use your detailed feedback in improving our paper and are happy to address any other questions you have. If our responses sufficiently address the weaknesses and questions that you raised, we kindly and respectfully ask if our rating could be improved.

---

> ### Author Response · Authors · 2023-08-16
> **More Intuition**
>
>
> We again thank the reviewer and will add more intuition for our lower bound proofs, which interestingly provide a separation between the $U = V$ and $U \neq V$ case. As mentioned in our submission, when $U \neq V$, we can reduce to the APX-Max-IP problem, where we want to find a small set of pairs of vectors that have a large inner product. Here the inner product is at least max/100 and there are at most $n^{o(1)}$ such pairs; see Assumption 4 in the submission.
>
> Consider this small set of large inner product pairs, which represents a small number of entries in $UV$ that are large in magnitude (i.e. when you apply a threshold at max/100, the resulting matrix is sparse). Since $f(x) = x^p$ amplifies the magnitude differences,  it follows that $f(UV)$ is much closer relatively to an approximately sparse, and therefore low rank, matrix (i.e. by thresholding and removing the terms that have magnitude less than $(max/100)^p$).  Therefore an approximate low rank approximation (LRA) algorithm can recover this sparse low rank matrix well enough so that the span of the approximate matrix can be used, via leverage score computations, to solve the APX-Max-IP problem. Our lower bounds do not generalize to the case when $U = V$ because the sparsity structure is broken as the diagonal of the matrix $UV$ is now larger than the maximum inner product of two different vectors, and this destroys the low rank structure. In some sense, the diagonal of our matrix is forced to include the dot product of $u_i$ with itself and this shifts the entire matrix by a large multiple of the identity, crucially removing the sparse + low-rank structure that we exploited in our lower bound argument before. Indeed, as mentioned in the paper, the positive semidefinite (PSD) structure when $U = V$ allows us to derive a fast LRA algorithm and our lower bound no longer holds in this setting.
>
> Furthermore, for our lower bounds for $f(x) = |x|^p$, note that we use a different structural property that holds even when $U = V$. For intuition as to why $p$ being odd is necessary for having a stronger lower bound in this setting, note that we now perform reduction to OVP. In this setting, note that our clever reduction forces the absolute value operation to essentially alter entries of $(UV)^p$ but only at entries of the original $UV$ with zero dot product. Therefore, we can write $f(UV)$ as a sum of a low rank matrix (from tensor product) and a sparse matrix, whose sparse entries now represent the OVP pairs. Then, either the sparse matrix forces LRA to have high error or if it does not have high error, then we can recover the sparse matrix by taking the difference of the LRA and the low rank tensor product. Note that this does not hold when p is even since the absolute value does not induce the additional sparse matrix; however, this structure still holds when $U = V$ since the diagonal terms are absorbed by the low rank matrix.
>
> If our understanding is correct, the reviewer does not seem to be concerned with the main contribution of our paper, which is our novel lower bounds for low rank approximation of element-wise transformed matrices. As the discussion phase will end soon, we ask if there is anything else we can help to clarify the contributions in our submission.

---

> > ### Comment · Reviewer_GCRp · 2023-08-16
> >
> > I thank the authors for addressing my concerns. The intuition for the having an odd p for a stronger lower bound is clearer now. I'm upgrading the score based on the responses. Please include the relevant points in the paper.

---

### Official Review · Reviewer_9ux2 · 2023-07-25

**Soundness:** 3 good
**Presentation:** 3 good
**Contribution:** 2 fair
**Rating:** 6
**Confidence:** 1

**Summary:**

For two matrices U,V one is often interested in computing f(UV) for some entrywise function f. Several papers in the past decade considered the problem of approximating f(UV) by a product AB with small inner-dimension (low-rank approximation). While a number of past papers presented clever positive results, this paper is about demonstrating various negative results. I am very far from the field of complexity and cannot evaluate soundedness of the claims. However, I should say that the negative results are probably best presented in a different venue (SODA etc), since this conference is mostly interested in doing things rather than in enumerating all things that cannot be done.

**Strengths:**

Key problem of contemp interest.

**Weaknesses:**

Complexity results are likely irrelevant to most attendees.

**Questions:**

n/a

---

> ### Author Rebuttal · Authors · 2023-08-08
>
> Thank you for your positive feedback. We would like to address your main comment concerning “irrelevant complexity results to most attendees” and “[NeurIPS] is mostly interested in doing things rather than in enumerating all things that cannot be done”:
>
> * First, while our submission mainly focuses on impossibility results, we note that we also provide relative and additive error low rank approximation algorithms that match our lower bounds of $n * 2^p$ and $n * \text{poly}(p)$, respectively.
>
> * Furthermore, all of our lower bounds are based on algorithmic reductions. We focus on the lower bounds as they present more novelty and difficulty than our upper bounds and in fact, stand out in a field that is dominated by positive results.
>
> * Given the large focus on Transformers currently, our paper’s resulting conclusions for the feasibility of efficient attention is very relevant and timely for NeurIPS.
>
> * More generally in the field of machine learning, we believe there have been multiple influential complexity (i.e., impossibility) results published at NeurIPS, to name a few out of many:
>   * Coarse Sample Complexity Bounds for Active Learning (Dasgupta, 2005)
>   * Sample Complexity of Testing the Manifold Hypothesis (Narayanan and Mitter, 2010)
>   * Sample Complexity of Episodic Fixed-Horizon Reinforcement Learning (Dann and Brunskill, 2015)
>   * Lower Bounds on the Robustness to Adversarial Perturbations (Peck et al, 2017)
>   * On the Complexity of Learning Neural Networks (Song, 2017)
>
> * Even more generally, we believe that lower bounds are important as they show the limitations of what is algorithmically possible, and so it makes sense for them to be presented along with upper bounds at the same venue for visibility, so that researchers know when to stop searching for better algorithms, without, e.g., making additional assumptions or changing the problem. We believe this viewpoint is shared by many at NeurIPS, and that the call for papers certainly includes such results.

---

> > ### Comment · Reviewer_9ux2 · 2023-08-21
> >
> > Thank you authors for your response. Considering the association of your result with transformers, I have updated my score to 6.

---

### Decision · Program_Chairs · 2023-09-21

**Decision:**

Accept (poster)

**Comment:**

The reviewers agree unanimously that the paper is solid and worthy of acceptance. Please incorporate reviewer feedback into the final camera-ready version.